# Cardiometabolic Risk Markers for Aboriginal and Torres Strait Islander Children and Youths: A Systematic Review of Data Quality and Population Prevalence

**DOI:** 10.3390/ijerph20136228

**Published:** 2023-06-26

**Authors:** Eamon O’Bryan, Christopher D. McKay, Sandra Eades, Lina Gubhaju, Odette Pearson, Jessica A. Kerr, Alex Brown, Peter S. Azzopardi

**Affiliations:** 1Global Adolescent Health Group, Maternal, Child and Adolescent Health Program, Burnet Institute, Melbourne, VIC 3010, Australia; 2Faculty of Medicine, Dentistry and Health Sciences, University of Melbourne, Melbourne, VIC 3010, Australia; 3Melbourne School of Population and Global Health, University of Melbourne, Melbourne, VIC 3010, Australia; 4Curtin Medical School, Curtin University, Perth, WA 6102, Australia; 5Aboriginal Health Equity Theme, South Australian Health and Medical Research Institute, Adelaide, SA 5000, Australia; 6Faculty of Health and Medical Sciences, University of Adelaide, Adelaide, SA 5000, Australia; 7Department of Psychological Medicine, University of Otago Christchurch, Christchurch 8011, New Zealand; 8Centre for Adolescent Health, Population Health Theme, Murdoch Children’s Research Institute, Parkville, VIC 3052, Australia; 9Department of Paediatrics, University of Melbourne, Parkville, VIC 3052, Australia; 10Telethon Kids Institute, Perth, WA 6009, Australia; 11National Centre for Indigenous Genomics, The John Curtin School of Medical Research, Australian National University, Canberra, ACT 2601, Australia

**Keywords:** Aboriginal and Torres Strait Islander, Australia, Indigenous, child, youth, cardiometabolic, disease risk, prevalence, metabolic syndrome, obesity, central adiposity, dyslipidaemia

## Abstract

Cardiovascular disease and type 2 diabetes mellitus are leading contributors to the health inequity experienced by Aboriginal and Torres Strait Islander peoples, and their antecedents can be identified from early childhood. We aimed to establish the quality of available data and the prevalence of cardiometabolic risk markers among Aboriginal and Torres Strait Islander children and youths (0–24-year-olds) to inform public health approaches. A systematic review of the peer-reviewed and grey literature was conducted between 1 January 2000–28 February 2021. Included studies reported population prevalence of cardiometabolic risks, including elevated blood pressure, obesity, central adiposity, dyslipidaemia, hyperglycaemia, and ‘metabolic syndrome’ for Aboriginal and Torres Strait Islander people aged 0–24 years. Fifteen studies provided population estimates. Data quality was limited by low response rates (10/15 studies) and suboptimal outcome measurements. Obesity is the most reported risk (13/15 studies). Aboriginal and Torres Strait Islander children have an excess risk of obesity from early childhood and prevalence increases with age: 32.1% of Aboriginal and Torres Strait Islander 18–24-year-olds had obesity and 50.8% had central adiposity. In a cohort of 486 9–14-year-olds in Darwin, 70% had ≥1 component of metabolic syndrome; 14% met the full criteria for the syndrome. The prevalence of cardiometabolic risk in Aboriginal and Torres Strait Islander young people is difficult to estimate due to limitations in measurement quality and sampling representativeness. Available data suggest that cardiometabolic risk markers are evident from early childhood. The establishment of national and state-level datasets and a core outcome set for cardiometabolic screening would provide opportunities for preventative action.

## 1. Introduction

Cardiovascular disease (CVD) and type 2 diabetes mellitus (T2DM) are now leading causes of disease globally and key contributors to the health inequity experienced by the Aboriginal and Torres Strait Islander peoples in Australia [1,2]. Traditionally diseases of adulthood, these diseases have an increasingly early age of onset; a recent cross-sectional study of Aboriginal and Torres Strait Islander youths under 24 years living in Northern Australia found an incidence of T2DM of 6.7 per 1000 people [3]. Importantly, these diseases can be averted by identifying and addressing key risks, particularly during childhood and youth [4,5].

Childhood and youth are defined by the United Nations as overlapping life stages ranging from 0–18 years and 15–24 years, respectively [6]. They are critical stages of physical, psychological and interpersonal development; health in this period lays the foundation for health into adulthood, just as the development of disease risk markers in childhood and youth can predict the development of disease later in life [7,8].

At least six anthropomorphic and serum biomarkers that are established risk markers for cardiometabolic disease can be reliably and objectively measured in clinical settings, as follows: obesity, as measured by elevated body mass index (BMI); central adiposity, as measured by elevated waist circumference (WC); elevated blood pressure (BP); raised serum triglycerides (TG); reduced high-density lipoprotein-C (HDL-C); hyperglycaemia [9,10,11]. Metabolic syndrome (MetS) is a diagnostic concept that clusters cardiometabolic risks as a means of predicting the cumulative risk of CVD and T2DM [12]. In a US-based longitudinal study, children diagnosed with MetS had 11 times the odds of developing T2DM as an adult [7]. Worldwide prevalence estimates for young people aged 0–19 years with MetS range from 2–23% depending on the criteria used, with up to 60% of young people who are overweight or obese meeting diagnostic criteria [13]. Table 1 summarises commonly used definitions of MetS for children and youths [14,15,16,17,18,19]. While these definitions are helpful to frame MetS in children and youths, it is important to note that several diagnostic criteria exist for each of the individual cardiometabolic risk factors that comprise MetS in young people aged 0–24 years.

Accurate prevalence estimates of cardiometabolic risk markers for Aboriginal and Torres Strait Islander young people at a population level would help inform age-appropriate screening and prevention programs [21]. For example, a group of experts previously recommended that screening for T2DM should be offered to any Aboriginal and Torres Strait Islander person over 10 years with a pre-disposing risk factor (any one of the following: overweight or obesity, a positive family history of diabetes, signs of insulin resistance, dyslipidaemia, or having received psychotropic therapy) [22]. In addition, a recent consensus statement recommended screening for cardiometabolic risk markers from age 18 years to identify young Aboriginal and Torres Strait Islander people with a high risk of a primary CVD event (GRADE good practice point) [23,24,25]. These recommendations, based largely on expert consensus, would benefit from understanding the population burden; it will help guide who should be screened, what should be screened, and which groups should be the priority. Several different criteria are used to define cardiometabolic risk markers, and there are currently no accepted standard criteria that define each of these risk markers in Aboriginal and Torres Strait Islander young people [24]. Given many of these cardiometabolic risk markers are asymptomatic, it is impossible to estimate their true prevalence without objective screening using gold-standard measurements at a population level.

This review aimed to establish the population prevalence for cardiometabolic risk markers in Aboriginal and Torres Strait Islander children and youths (0–24-year-olds, as defined by the United Nations) across Australia to inform public health approaches [6]. We also reviewed the coverage and quality of these estimates to inform further investments in data collection and surveillance.

## 2. Materials and Methods

The protocol for this systematic review was registered with the International Prospective Register of Systematic Reviews (PROSPERO, CRD42020153930) on 28 April 2020.

We systematically searched the literature from 1 January 2000–28 February 2021. The search was conducted on Medline, Embase, PsychINFO, Scopus, CINAHL, and Informit. The final search terms are included in Appendix B Table A1. The grey literature (research not published commercially) from the Australian Institute of Health and Welfare (AIHW), Australian Bureau of Statistics (ABS) and Australian Indigenous HealthInfoNet was hand-searched, as were reference lists from published reviews and meta-analyses to identify additional articles of relevance.

A title and abstract screen, and a two-stage (stage I and stage II) full-text review were performed by two independent reviewers (E.O. and C.D.M.) and reported using PRISMA guidelines (Figure 1) [26]. In stage I, quantitative studies reporting prevalence data for cardiometabolic risk markers (obesity defined by elevated BMI, central adiposity defined by elevated WC, elevated BP, elevated serum TG or reduced HDL-C, hyperglycaemia, MetS) specific to Aboriginal and Torres Strait Islander children and youths (aged 0–24 years) were included (that either sampled Aboriginal and Torres Strait Islander young people exclusively or reported disaggregated estimates as part of a larger study). A stage II full-text review was then undertaken with a focus on sample quality (representative at a population level or from a substantial longitudinal study; substantial longitudinal cohorts were defined as those that followed up with a cohort of ≥60 individuals through childhood and youth) and outcome measure quality (relative to the gold-standard for a given outcome). Excluded articles were reviewed, with conflicts between the two reviewers discussed and resolved with the help of a third independent reviewer (P.S.A). For the included articles, we extracted data and undertook a quality assessment. Inclusion and exclusion criteria for stage I and II of study selection are depicted in Table 2.

The following data were extracted from each study: study characteristics (title, author, year published, journal or database of article, location of study, study period, study design), participant characteristics (number of participants, age range and mean age, sex distribution and proportion of children and youths, ethnicity of population and proportion of Aboriginal and Torres Strait Islander participants), outcome measures (prevalence of MetS, elevated BP, dyslipidaemia defined as low HDL-c or elevated TG, obesity defined as elevated BMI, central adiposity defined by elevated WC, and hyperglycaemia) and the criteria used to define the prevalence of these outcomes.

Study quality was assessed using two independent assessment tools. The SAHMRI Centre of Research Excellence in Aboriginal Chronic Disease Knowledge Translation and Exchange (CREATE) Quality Appraisal Tool was selected to examine study quality as it pertained to Aboriginal and Torres Strait Islander involvement, governance, and cultural recognition and respect [27]. The Joanna Briggs Institute (JBI) Critical Appraisal Checklist for prevalence studies was undertaken to assess the methodological quality of the included studies [28].

Included studies were grouped by the outcome of interest, the age of the cohort (defined as preschool children 0–5 years, children 6–14 years, and youths 15–24 years), and the corresponding population (nationally representative, representative of a state or community, or those with a substantial longitudinal cohort). The studies were analysed by descriptive synthesis given the breadth of study cohorts and outcomes and the variability of criteria used for each risk factor. A meta-analysis was not performed due to the heterogeneity of the studies.

## 3. Results

The systematic literature search identified 1762 articles of potential relevance, with 1302 excluded after the title and abstract screening (Figure 1). A total of 460 articles underwent stage I full-text analysis, with 43 articles selected for stage II full-text analysis, and 21 articles (from 15 distinct studies) included for data extraction (Figure 1). Included studies ranged in sample size from 78 to 4763 (Appendix A). Of the 15 studies, 13 reported the prevalence of obesity (elevated BMI), 6 reported hyperglycaemia, elevated HbA1c, or known T2DM, 4 reported prevalence of elevated BP, 5 reported elevated WC, 3 reported elevated TG or low HDL-c, and 2 reported prevalence of MetS. Studies that were excluded after Stage II were listed in Table A2 with reason for exclusion noted.

All included studies scored ≥ 5 (out of a possible 9) in the JBI Critical Assessment Tool for prevalence studies, with all describing the study population and setting in detail and including a sufficient sample size (*n* ≥ 60 participants) [28]. Furthermore, 10 of the 15 studies had low response rates, and 8/15 had a low proportion of Aboriginal and Torres Strait Islander participants. Some outcome measures were also suboptimal: in all seven studies that reported the prevalence of elevated BP, it was measured at a single timepoint (two time-points is the accepted standard) [29], and the two studies that reported state-wide incidence of T2DM using a retrospective chart audit did not account for asymptomatic or undiagnosed cases. Table A3 outlines the number of studies that met each item of the JBI Critical Assessment Tool.

Study quality relating to the CREATE quality appraisal tool was generally low: 4/15 studies mentioned community consultation and engagement prior to the commencement of the study, and 8/15 studies stated that they followed and respected local community protocols [27,28]. Table A4 outlines the number of studies that met each item of the CREATE quality appraisal tool.

Prevalence estimates are summarised in Table 3 for preschool children (0–5 years), children (6–14 years), and youths (15–24 years), with most estimates being for children (*n* = 13).

### 3.1. Preschool (0–5 Years)

#### Obesity (Elevated BMI)

Studies involving preschool children (*n* = 5) exclusively reported obesity prevalence based on BMI cut-points defined by International Obesity Taskforce (IOTF) and World Health Organisation (WHO). The prevalence of obesity among this age group ranged from 5.8% to 22.1%, including national and regional estimates [30,31]. Available data were not disaggregated by sex.

The national-level 2018/2019 National Aboriginal and Torres Strait Islander Health Survey (NATSIHS) by the Australian Bureau of Statistics (ABS) reported that 10.8% of Aboriginal and Torres Strait Islander 2–3-year-olds were categorised as obese (IOTF), up from 6% among 2–4 year-olds in the 2012/2013 ABS survey [32,33]. In comparison, 5.1% of non-Indigenous 2–4-year-olds in the 2012/2013 comparator survey had obesity [32,33]. In the national Longitudinal Study of Indigenous Children (LSIC) using age-standardised cut-points (WHO), prevalence ranged from 3–14% in the younger cohort between the ages of 0 and 5 years, while in the older cohort prevalence was 8% at mean age 3.5–5.5 years [34].

Regional estimates of obesity in preschool children were available from two studies: 22.1% of a cohort of 24-month-old children living in Western Sydney in New South Wales (NSW) had obesity as defined by Centre for Disease Control and Prevention (CDC) z-score cut-offs, while a school-based study in the Hunter New England region in NSW reported that 5.8% of 2–5-year-old children had obesity using IOTF cut-offs [30,31].

### 3.2. Children (6–14 Years)

#### 3.2.1. Obesity (Elevated BMI)

Obesity was the most common outcome measure for children aged 6–14 years (9/11 studies) [32,33]. Prevalence of obesity in this age group ranged from 2.9% to 18.6%, including national and regional estimates [35,36]. These studies used IOTF and CDC z-score cut-offs to define obesity [37,38].

National-level ABS data demonstrated a slight increase in obesity over time, from 11.8% among 10–14-year-olds in 2012/2013 to 13.8% among 9–11-year-olds in 2018/2019 [32,33]. Among LSIC’s younger cohort, obesity prevalence was 13.5% at a mean age of 5.5–6.5 years [39]. Among the older cohort, who were followed longitudinally during this stage of life, obesity increased from 9.1% at a mean age of 5.5–6.5 years to 18.2% at 8.5–9.5 years [39].

The ABC study provided estimates of Aboriginal and Torres Strait Islander children with obesity born in Darwin, Northern Territory (NT); however, two papers used different criteria to define obesity, resulting in different prevalence estimates between each paper. Sjoholm et al. identified that 2.9% of 9–13-year-old participants had obesity as defined by IOTF, while Sellers et al. reported a prevalence of 4.9% among 9–14-year-old participants living in Darwin using Centres for Disease Control (CDC) z-score cut-offs to define obesity [35,40]. Data sampled from urban and rural schools in two states or territories also suggest an increase in obesity by age: a study of 5–7-year-old schoolchildren in the ACT reported that 7% of Aboriginal and Torres Strait Islander participants had obesity (IOTF), while a study of schoolchildren in NSW reported 10.4% obesity prevalence among 5–16-year-old Aboriginal and Torres Strait Islander children (IOTF) [41,42]. Another study found that 15% of 5–17-year-olds attending one of three participating schools in the remote Torres Strait Islands had obesity [43].

The 2012/2013 and 2018/2019 ABS surveys disaggregated national data for children aged 2–14 years by sex [32]. In both studies, obesity prevalence was similar among females and males in this age group (10.6% of females and 9.8% of males in 2012/2013, and 13.3% of females and 12.3% of males in 2018/2019) [32,33].

#### 3.2.2. Central Adiposity (Elevated WC)

The prevalence of central adiposity in children was reported in two studies [40,43]. Both studies reporting the prevalence of central adiposity were located in Northern Australia. In the Aboriginal Birth Cohort (ABC) in the Northern Territory, 26.3% of 9–14-year-olds (*n* = 489) had central adiposity (WC > 90th percentile for age and sex) [40]. In the Torres Strait school study, the prevalence was 38% (WC ≥ 94 cm in males, ≥80 cm in females) [18,20,43]. Within this study, central adiposity was more prevalent in females (59%) relative to males (13%) [43].

#### 3.2.3. Elevated BP

Elevated BP prevalence data were reported in three studies, each of which measured BP at a single time-point. The definition of elevated BP varied between studies; BP ≥ 90th percentile for age and sex compared with BP ≥ 95th percentile for age and sex [36,40,43]. Prevalence estimates for this age group ranged from 7.3% to 40%. There were no prevalence data disaggregated by sex.

The Antecedents of Renal Disease in Aboriginal Children (ARDAC) study (baseline *n* = 1949) identified an increase in the prevalence of elevated BP over time in urban, rural, and remote NSW primary school children: 7.3% of the cohort had elevated BP (BP ≥ 95th percentile) at a mean age of 10.5 years, compared with 9.1% at a mean age of 14.5 years [36]. The ABC study provided estimates of elevated BP for a subset of participants with normal BMI but elevated WC. Within this group, 40% of 9–14-year-old participants had elevated BP (BP ≥ 90th percentile) [40]. In the Torres Strait school study, 27% of 5–17-year-olds had elevated BP [43].

#### 3.2.4. Hyperglycaemia

Data were available for the prevalence of pre-diabetes in children in two studies, both of which used criterion FBG ≥ 5.6 mmol/L. Two additional studies provided state-wide incidence estimates over time for 0–18-year-old children, collected by chart audit of diagnosed T2DM using the American Diabetes Association criteria for T2DM [44]. Pre-diabetes prevalence estimates ranged from 2% to 12%, while diabetes incidence ranged between 12.6/100,000 person-years to 31.1/100,000 person-years. Available data were not disaggregated by sex.

State-wide incidence data for diagnosed T2DM were available for Western Australia (WA) and NSW. In WA between 1990–2012, the mean annual incidence of Aboriginal and Torres Strait Islander children under 18 years diagnosed with T2DM was 12.6/100,000 person-years, increasing from 4.5/100,000 person-years in 1990 to 31.1/100,000 person-years in 2012 [45]. Using a retrospective audit of diagnosed cases of T2DM, the NSW study reported a mean annual incidence of 3.1/100,000 person-years between 2001–2008 [46].

Prevalence of pre-diabetes (FBG ≥ 5.6 mmol/L) was reported for Aboriginal children aged 9–14 years born in urban Darwin, with 2% of participants meeting this criterion [40]. Using the same criterion, 12% of 5–17-year-old children participating in the Torres Strait Island school study who were overweight or obese also had pre-diabetes [38,43].

#### 3.2.5. Metabolic Syndrome (MetS)

Prevalence estimates for MetS were available from two studies [40,43,47]. Juonala [40] and Sellers [47] both utilised data from the ABC study, but defined MetS differently using National Cholesterol Education Program (NCEP) III and Modified NCEP (MetSNCEP75), respectively (Table 1). The Torres Strait Island school study used the IDF criteria to define MetS.

Prevalence estimates for MetS were only available from the Darwin region of the NT and the Torres Strait Islands. In the Torres Strait study, 17% of 5–17-year-olds met the criteria for MetS (IDF criteria), while between 14% (MetSNCEP75) and 15.7% (NCEP III) of 9–14-year-olds in ABC had MetS [40,43,47]. Within ABC, 70% of study participants had at least one risk marker by age 14 [40]. Both prevalence estimates were limited by sampling representativeness: ABC comprised a non-random cohort of children born to Aboriginal mothers at Royal Darwin Hospital, while MetS could only be assessed for 61.4% of the Torres Strait cohort, due to low uptake rates for blood sampling [40,47].

### 3.3. Youth (15–24 Years)

#### 3.3.1. Obesity (Elevated BMI)

Prevalence estimates of obesity in Aboriginal and Torres Strait Islander youths were reported by six studies, all of which utilised IOTF criteria. Prevalence estimates for obesity in this age group ranged from 6.4–31.1%.

From the 2018/2019 ABS survey, 18.3% of 15–17-year-olds had obesity, compared with 31.1% of 18–24-year-olds. However, the accuracy of the estimates might be impacted by low response rates in both age groups (53.6% and 34.0%, respectively) [32]. Prevalence was lower in the earlier 2012/2013 survey: 14.2% of 15–17-year-olds and 28.4% of 18–24-year-olds. Non-Indigenous comparison estimates were not provided in 2018/2019, though in 2012/2013 the prevalence of obesity among non-Indigenous peers was 7.4% for 15–17-year-olds and 14.4% for 18–24-year-olds [33,38]. Nation-wide cross-sectional data from the school-based National Youth Cultures of Eating Study also identified an increase in obesity in 13–18-year-old participants between 2006 and 2012, from 6.5% to 7.0, respectively% [48].

Longitudinal data from urban, rural, and remote schools in NSW reported an obesity prevalence of 21.5% at a mean age of 16.5 years and 19.2% at 18.5 years [40], while the ABC study in NT reported that 6.4% of 16–20-year-olds had obesity (up from 2.9% at 9–13 years) [35].

The 2018/2019 ABS survey disaggregated prevalence data for obesity by sex for 15–24-year-olds. Obesity rates were similar for males and females: 18.3% of 15–17-year-old males and 18.1% of females had obesity, while 31.1% of 18–24-year-old males and 33.1% of 18–24-year-old females had obesity [32].

#### 3.3.2. Obesity (Increased WC)

Disaggregated prevalence data for central adiposity (elevated WC) in youth were available from two studies, both of which utilised WHO criteria for central adiposity (WC ≥ 94 cm in males and ≥80 cm in females) [49].

Prevalence estimates were available for the 2012/2013 and 2018/2019 national ABS studies [32,33]. Half (50.8%) of 18–24-year-old youths recruited into the 2018/2019 ABS survey had central adiposity (elevated WC) using WHO criteria [32]. Central adiposity was more prevalent among females (62.9% of females, 39.9% of males). These estimates were comparable to earlier estimates from the 2012/2013 AATSIHS using the same criteria, in which 39.6% of 18–24-year-old Aboriginal and Torres Strait Islander males and 67.2% of females had increased WC [33,49].

#### 3.3.3. Elevated BP

Data for elevated BP in youth were available from five studies. The 2012/2013 and 2018/2019 ABS surveys, and Wang [50] study of a remote NT community defined elevated BP as BP ≥ 140/90 mmHg, while two longitudinal studies (the ARDAC study and the Study on Environment on Aboriginal Resilience and Child Health, or SEARCH) used BP ≥ 95th percentile for age and sex [32,33,36,50,51]. Prevalence estimates for elevated BP ranged from 8.8–18.3% [32,36].

The prevalence of elevated BP (≥140/90 mmHg) for 18–24-year-olds in the national 2018/2019 ABS survey was 8.8% [32]. In the ARDAC study of schoolchildren in NSW, 18.3% of youths with median age of 18.5 years had elevated BP (≥95th percentile for age and sex) [36]. Data from a matched pair study in a remote Northern Territory (NT) community demonstrated a decrease in the prevalence of elevated BP (≥140/90 mmHg) between 1992–1997 and 2004–2006, from 11.5% to 3.6% in males and from 7.0% to 0.9% in females [50].

#### 3.3.4. Dyslipidaemia (Elevated TG or Reduced HDL-C)

Dyslipidaemia (TG ≥ 2.0 mmol/L, or HDL-C < 1.0 mmol/L in males or <1.3 mmol/L in females) prevalence was reported in the 2012/2013 ABS survey: 16.4% of 18–24-year-old study participants had elevated TG, while 29.5% had reduced HDL-C [33]. Data were not disaggregated by sex.

#### 3.3.5. Hyperglycaemia

National-level prevalence data for pre-diabetes and diabetes in youths were reported in the 2012/2013 ABS survey. Pre-diabetes was prevalent among 2.4% of 18–24-year-olds in the study. Diabetes was defined using both FBG and HbA1c measures. Using HbA1c ≥ 6.5%, 0.4% of youths had diabetes (type 1 or type 2 diabetes mellitus), compared with 0.1% using the gold-standard measure of FBG ≥ 7.0 mmol/L [33]. Data were not disaggregated for T1DM or T2DM, so some T1DM may have been included in this estimate. Data relating to the prevalence of diabetes were not disaggregated by sex.

## 4. Discussion

To our knowledge, this is the first systematic review to investigate the prevalence of cardiometabolic risk markers in Aboriginal and Torres Strait Islander young people. This review builds on the findings of a previous publication by our team investigating potential determinants of cardiometabolic risk in this population group [52]. Our findings suggest that Aboriginal and Torres Strait Islander children have a substantial risk of obesity from early childhood and that the prevalence of obesity increases from childhood into youth, from 10.8% among 2–4-year-olds, compared to 32.1% among 18–24-year-olds, and prevalence of central adiposity of 50.8% among the older age group [32]. Few studies screened for pre-diabetes or T2DM; however, state-based hospital audits of Aboriginal and Torres Strait Islander children under 18 with diagnosed T2DM identified annual incidence rates of up to 12.6 per 100,000 person-years [42]. The limited data relating to elevated BP and dyslipidaemia suggest that these are prevalent in youth; 8.8% of 18–24-year-olds nationally had elevated BP and 16.4% had raised serum TG [32,33]. Limited prevalence data for MetS (clustering of risk markers) indicate that this syndrome may also be common, with 14% among a Darwin cohort, with 70% having at least one risk marker, and 17% among a cohort of Torres Strait school children [40].

Available data highlight differences in cardiometabolic risk burden by sex, across age groups, and between metropolitan, regional, and remote Australia. Females were more likely to have central adiposity; 62.6% of 18–24-year-old females nationally had central adiposity compared with 39.9% of males [43]. Conversely, males carried a marginally increased risk of elevated BP [32,50]. In a study in a remote Aboriginal community, females were more likely to have obesity compared with males (6.3% compared with 4.5%) [50]. Obesity appears well established in early childhood and, along with overall cardiometabolic risk, continues to increase from childhood to youth. Longitudinal data from public primary schools in NSW demonstrated an increase in the prevalence of elevated BP from 7.3% at baseline to 18.3% after 8 years’ follow-up, alongside the increase in obesity prevalence from childhood to youth [36]. Elevated cardiometabolic risk was observed Australia-wide; however, inconsistencies in prevalence estimates from remote communities suggest the need for further primary care-based research in these areas, as 8% of normal-weight children and 12% of overweight or obese children demonstrated impaired FBG in a Torres Strait community, compared with 2% in the ABC [43,50]. These discrepancies highlight the importance of community-centred research to better understand the diverse health needs of young people living in remote Australia.

The true burden of cardiometabolic disease risk in Aboriginal and Torres Strait Islander young people is difficult to estimate due to limitations in measurement consistency and quality, in sampling representativeness, and in data disaggregation. All studies investigating elevated BP (*n* = 7) did so from measures taken at a single time-point, which decreases the accuracy of results: in a study of US adults, only 3% of people with BP ≥ 140/90 mmHg on a single measurement had BP ≥ 140/90 mmHg on an average of three sequential measurements [32,53,54]. The 2012/2013 ABS study’s national estimate of 18–24-year-olds with diabetes did not disaggregate between type 1 and type 2 diabetes [33]. Furthermore, the two state-based studies that reported the incidence of T2DM through retrospective chart audits may have underestimated incidence, as they were unable to quantify the burden of asymptomatic or non-presenting T2DM among Aboriginal and Torres Strait Islander children and youths [45,46,55]. The differences between prevalence rates of obesity (elevated BMI) within the same cohort using different criteria (IOTF, WHO, and CDC BMI z-score cut-points were variably used to define obesity) highlight the need for standardised criteria to define risk markers in Aboriginal and Torres Strait Islander young people [35,40]. Furthermore, the high prevalence of elevated WC in children with normal BMI demonstrates the potential limitations of BMI as a measure of metabolic dysfunction in Aboriginal and Torres Strait Islander people [40,43]. Children diagnosed with MetS in the Darwin cohort had a low mean BMI z-score of 0.67 (95% CI 0.22, 1.13); however, the mean WC z-score was 2.69 (1.91, 3.47), well above the 95th percentile for age and sex [40,43]. Similarly, while 15% of Torres Strait Islander schoolchildren met the criterion for obesity, 38% were diagnosed with central adiposity [43]. This supports previous studies that indicate that increased WC is a more sensitive measure of metabolic risk than BMI in Aboriginal Australians, as it identifies individuals with increased visceral adipose tissue associated with the development of T2DM [56,57,58,59,60]. Few studies reported the prevalence of hyperglycaemia or dyslipidaemia, likely due to difficulty in obtaining serum samples for a large study population and in young age groups, with the 2012–2013 AATSIHS being the only study to provide national-level data for these cardiometabolic risk markers. Within this study, only participants aged 18–24 years were included. The participation rate of only 40.4% within this age group reduces the reliability of these results as accurate national estimates [33]. Data reported were often not disaggregated by sex, particularly in studies that reported the prevalence of obesity in preschool children (0–5 years) and children aged 6–14 years [30,31,34,36,41]. Similarly, state-wide incidence data for T2DM in Aboriginal young people in WA and NSW were not disaggregated by sex [45,46]. Without this data, interpretation of differences in cardiometabolic risk between sexes was limited. Although good quality data were available for estimates of obesity, reliably measured, representative prevalence estimates for other cardiometabolic risk markers during this period were lacking. Despite this, available data for these other cardiometabolic risk markers suggest a high burden of risk from childhood and reinforce the need for improved screening in younger age groups.

Azzopardi et al.’s [61] systematic review of the quality of health research for Aboriginal and Torres Strait Islander young people aged 10–24 years highlighted several gaps in the available data relating to disease burden in this age group. Of the 240 studies identified in the study as being of good quality, most focused on communicable disease, oral health, and substance use, while 10 studies investigated cardiometabolic risk markers, such as obesity, high blood pressure, and high cholesterol. Accurate estimates of cardiovascular and metabolic disease burden in 10–24-year-olds were not available due to limitations in data disaggregated by age [61]. Dyer et al.’s systematic review of overweight individuals and obesity in Aboriginal and Torres Strait Islander children identified a similar range of studies investigating the prevalence of obesity, and estimated an obesity prevalence of 7.8–10.5% in children <18 years [62]. We excluded a number of studies included in Dyer et al.’s review during our data extraction stage due to poor representativeness and issues with the sampling framework. Eight studies included in this review were published after Dyer et al.’s review in 2017, including the most recent 2018/2019 NATSIHS. These more recent studies suggest that the prevalence of obesity in this age group is substantially greater than previous estimates [32,34,39]. Comparison with other First Nations communities indicates that increased childhood cardiometabolic risk is a global phenomenon: 14.1% of Alaskan Native high school students had obesity in 2015, while a retrospective analysis of Māori children in Auckland reported a T2DM incidence of 3.4/100,000 person-years, similar to the 3.0/100,000 person-years identified in Aboriginal and Torres Strait Islander children in NSW. Available data for non-Indigenous children highlights the disparity in cardiometabolic risk between Indigenous and non-Indigenous children worldwide: Haynes et al.’s Western Australian study reported 0.6/100,000 person-years in non-Indigenous children, while the same New Zealand study reported mean annual incidence 0.1/100,000 person-years among ethnically European New Zealand children [45,46,63,64].

There are manifold contributors to the high rates of cardiometabolic risk in Aboriginal and Torres Strait Islander children and youths, many of them driven by Australia’s history of violent colonisation and racist government policy [65]. Available data indicate that Aboriginal and Torres Strait Islander young people carry a greater burden of cardiometabolic risk than their non-Indigenous peers. This was observed in prevalence rates of obesity (14.2% compared with 7.4% in 15–17-year-olds) and in the incidence of T2DM (incidence risk ratio 6.1 for Aboriginal and Torres Strait Islander children < 18 years compared with non-Indigenous counterparts) [33,55]. Aboriginal and Torres Strait Islander women have a prevalence odds ratio of 3.63 (95% CI 2.35–5.62) of developing gestational diabetes in pregnancy compared with non-Indigenous women, increasing the risk of intrauterine and post-pregnancy complications for woman and child [66,67]. Intergenerational poverty experienced by many Aboriginal and Torres Strait Islander families contributes to high rates of metabolic dysfunction through psychological stressors and poor diet [68,69,70]. In the LSIC, children whose BMI z-score increased at 3-year follow-up were more likely to live in an environment with high financial stress (30.7% vs. 27.4%) and poor diet (33.3% vs. 26.9%) [34]. However, a recent review of the potential determinants of cardiometabolic risk among Aboriginal and Torres Strait Islander children and youths by McKay et al. found that evidence is largely lacking for this population group, and further research is needed, particularly looking at social and environmental factors [52].

This review utilised a robust study protocol and involved an independent second reviewer to minimise selection bias. The grey literature search was thorough and identified important government and community-based reports which were synthesised in the final review. Extracted studies were assessed using two critical appraisal tools to objectively examine data quality and risk of bias. We were limited by the heterogeneity of study populations, outcomes, and the criteria used to define each outcome. This precluded meta-analysis and made inter-study comparisons difficult. This was mitigated by grouping studies by the scope of their population into nationally representative studies, community- or state-representative studies, and studies involving a substantial longitudinal cohort. This allowed inferences around prevalence estimates to be made, particularly regarding nation-wide studies. Assessment of a study’s cultural competency using the CREATE tool was hindered by a lack of preferred reporting guidance for Aboriginal and Torres Strait Islander research; studies may not have reported methodology undertaken that would meet CREATE criteria for culturally competent Aboriginal and Torres Strait Islander health research.

The evidence synthesised in this review highlights some important considerations for the assessment of cardiometabolic risk and areas where data can be improved. Findings from this review suggest that MetS is limited as a sensitive measure of early cardiometabolic risk and that understanding a young person’s risk requires earlier screening of individual risk markers. While several good quality studies were identified, we identified the need for greater investment in risk factor profiling and screening. While three studies achieved estimated state-wide coverage rates of >99% of diagnosed cases of T2DM, hyperglycaemia is often asymptomatic and, therefore, may be undiagnosed without routine screening in a paediatric population [45,46,55]. Although national and state-level datasets may not adequately represent all Aboriginal and Torres Strait Islander young people, and do not reflect the remarkable diversity of cultures within that population, greater investment in representative studies with strong community engagement and further longitudinal studies, following LSIC and ABC, with anthropomorphic and biochemical risk factor profiling, would provide a clearer directive for national and state health policy decisions [32,34,40].

Given the burden of cardiometabolic risk in children under 18 years, and the gaps in high-quality data from remote communities, this review supports providing cardiometabolic risk screening during early childhood for the Aboriginal and Torres Strait Islander peoples. At present, the Aboriginal and Torres Strait Islander Health Check (Medicare Item 715) does not offer screening for Aboriginal and Torres Strait Islander young people for elevated blood pressure until the age of 12; testing for elevated WC and biochemical risk markers, such as hyperglycaemia and dyslipidaemia, is not funded until a minimum of 18 years [71]. The establishment of a core outcome set for estimating cardiometabolic risk in Aboriginal and Torres Strait Islander young people, whereby a minimum set of risk markers are screened for using consistent measurement and reporting to determine an individual’s cardiometabolic risk, would improve available data quality and facilitate opportunities to treat and manage these risk markers to prevent the development of CVD and T2DM [72]. From the data identified in this review, several of these cardiometabolic risk markers are highly prevalent before the age of 18. Given the limitations of MetS in identifying early cardiometabolic risk, an appropriate core outcome set would include obesity, as measured by elevated body mass index (BMI) and elevated waist circumference (WC), elevated blood pressure (BP), raised serum triglycerides (TG), reduced high-density lipoprotein-C (HDL-C), and hyperglycaemia. Earlier screening with the establishment of a standardised core outcome set would provide a window in time in which to optimise health rather than anticipate illness.

## 5. Conclusions

Estimates of the cardiometabolic risk profile of Aboriginal and Torres Strait Islander young people are limited by data quality, but the available data signal that elevated cardiometabolic risk begins in early childhood. Greater investment in national and state-level datasets and the establishment of a standardised core outcome set for earlier primary care screening of cardiometabolic risk markers before the age of 18 would enable greater understanding of this risk burden. Furthermore, by identifying cardiometabolic risk at a younger age, young people would have greater opportunity to work with primary health providers to improve their health and reduce cardiometabolic risk prior to adulthood.

## Figures and Tables

**Figure 1 ijerph-20-06228-f001:**
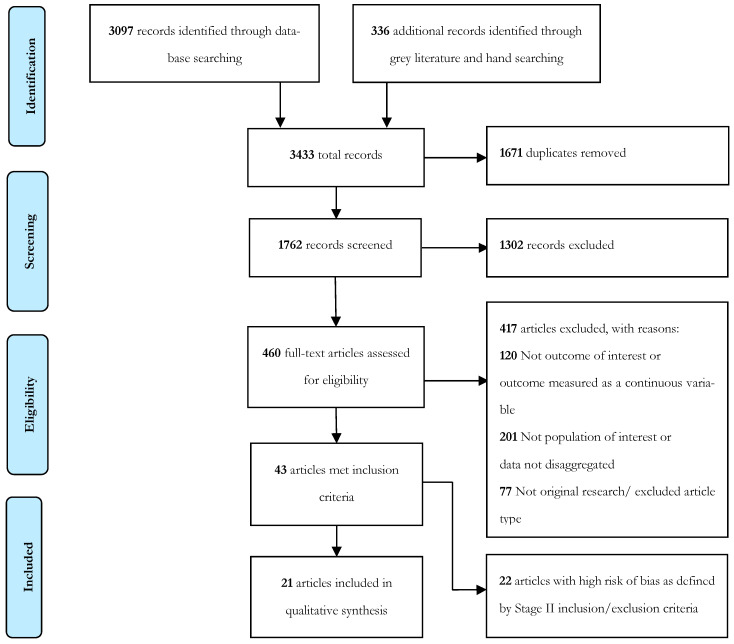
The search and study selection process depicted as a PRISMA flowchart, with finalised Medline search terms.

**Table 1 ijerph-20-06228-t001:** Common criteria used to define metabolic syndrome (MetS) in children and youths.

National Cholesterol Education Program (NCEP) Adult Treatment Panel (ATP) III (12–19-Year-Olds) [18]	International Diabetes Federation (IDF) (10–16-Year-Olds) [20]	Modified NCEP (MetSNCEP75) (3–18-Year-Olds) [8]	IDF (≥16-Year-Olds, Adult Criteria) [20]
≥3 of the following 5 components: - TG ≥ 1.24 mmol/L; - HDL-c ≤ 1.03 mmol/L; - Waist circumference > 90th percentile for age and sex; - FBG ≥ 6.1 mmol/L (or ≥5.6 mmol/L for impaired fasting glucose (IFG)); Systolic or diastolic BP > 90th percentile for age, height, and sex.	MetS diagnosed if child is ≥10 years of age, has central obesity (WC ≥ 90th percentile in males, ≥80th percentile in females) and ≥2 of the following: - Elevated TG ≥ 1.7 mmol/L; - Low HDL-C < 1.03 mmol/L in men, <1.30 mmol/L in women; - High blood pressure ≥ 130/85 mmHg; - Increased fasting plasma glucose ≥ 5.6 mmol/L.	MetS diagnosed if child has ≥3 of the following 5 components: - WC ≥ 75th percentile; - BP ≥ 75th percentile; - HDL-C ≤ 25th percentile; - TG ≥ 75th percentile; - Glucose ≥ 75th percentile.	MetS diagnosed if youth is ≥16 years of age, has central obesity (ethnic-specific values provided for different ethnic groups) and ≥2 of the following: - Elevated TG ≥ 1.7 mmol/L; - Low HDL-C < 1.03 mmol/L in men, <1.30 mmol/L in women; - High blood pressure ≥ 130/85 mmHg; Increased fasting plasma glucose ≥ 5.6 mmol/L.

**Table 2 ijerph-20-06228-t002:** Inclusion and exclusion criteria for stage I and stage II of study selection.

Stage I inclusion and exclusion criteria applied to all full-text studies remaining after the title and abstract screening (460 studies).	
Inclusion: articles will be included if they meet all of the following criteria	Exclusion: articles will be excluded if they meet any of the following criteria
Include Aboriginal and/or Torres Strait Islander participants living in Australia Participants are aged 0–24 years Report quantitative data on the prevalence of metabolic syndrome or its components Original published research: primary data English language	Data cannot be extracted separately for the population of interest Study only recruited participants with a specific pre-existing congenital or genetic condition, or an acquired condition not directly related to MetS Study is not original (including conference proceedings, posters or abstracts, editorials, commentaries, perspectives, book chapters, dissertations, meta-analyses and other systematic reviews) Abstract or full-text not available, or methodology missing or incomplete Animal study
Stage II inclusion and exclusion criteria applied to all studies remaining after the full-text review (42 studies).	
Inclusion: articles will be included if they meet all of the following criteria	Exclusion: articles will be excluded if they meet any of the following criteria
Study cohort is either: Nationally representative State representative Representative of a community A substantial longitudinal cohort study Study used the gold-standard measurement tool, e.g., fasting blood glucose	Participation rate of study cohort <60% Prevalence data reported for a study cohort that is not representative of the population of interest at a national, state, or community level, such as: Non-random sampling Small sample size (*n* < 60) Poor coverage of community Data are only reported as a mean value rather than a prevalence figure Used an unreliable or non-verified measurement tool to report the prevalence of a risk factor, e.g., finger-prick glucose

**Table 3 ijerph-20-06228-t003:** Overall prevalence estimates for outcomes by age groups.

	Outcome Prevalence					
Age Group	Obesity (Increased BMI)	Elevated WC	Elevated BP	Elevated TG; Reduced HDL-C	Hyperglycaemia or T2DM	Metabolic Syndrome
Preschool (0–5 years)	5.8–22.8%	N/A	N/A	N/A	N/A	N/A
Children (6–14 years)	2.9–18.6%	26.3–38%	7.3–27%	N/A	Pre-diabetes 2%, incidence T2DM 3.1–12.6 per 100,000 person-years	14–17%
Youth (15–24 years)	6.4–31.1%	50.8%	8.8–18.3%	Elevated TG 16.4%, reduced HDL-C 29.5%	Pre-diabetes 2.4%, diabetes 0.4% (HbA1c) or 0.1% (FBG)	N/A

## Data Availability

Not applicable.

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
