# Peer review of "Cardiometabolic Risk Markers for Aboriginal and Torres Strait Islander Children and Youths: A Systematic Review of Data Quality and Population Prevalence"

_ijerph, 2023, doi:10.3390/ijerph20136228_

Round 1

Reviewer 1 Report

This study offers valuable insights and actionable guidance for a higher risk population. How does it differ from the following manuscript?

McKay CD, O'Bryan E, Gubhaju L, McNamara B, Gibberd AJ, Azzopardi P, Eades S. Potential Determinants of Cardio-Metabolic Risk among Aboriginal and Torres Strait Islander Children and Adolescents: A Systematic Review. Int J Environ Res Public Health. 2022 Jul 27;19(15):9180.

The protocol followed is scientifically sound; however, overall, the manuscript needs more thoughtful introspection. For example, the reasons for setting an age range of 0 to 24 years must be discussed and supported with evidence. For example, no biochemical indices are provided for children 0 to <3 years and the age range for children falling into the adult parameters needs to be defined. See details below with more thoughts on these comments.  Inclusion of the directions in the Results and Discussion sections suggests that the authors need to review the manuscript more carefully before resubmission.

Line 46: Living in the Top End—please provide more information on this location for those who live outside of Australia.

Line 83: Change to past tense, e.g., we also reviewed…

Figure 1:

·       “3433 total records from” Delete “from.”

·       417 articles excluded, with reasons—needs reformatting, reason going below box and not visible.

·       44 articles included in. Correct 44 to 21 (42-22 = 21). Also fix formatting so word after “in” is visible.

Recommend putting Figure 1 after inclusion/exclusion criteria discussion and Figure 2.

Figure 2:

·       Stage II Inclusion, remove semi-colons at end of items to be consistent with other boxes in the figure.

·       Also, be consistent with period usage. Either use throughout or eliminate from the few criteria that end in a period.

Table 1:

·       Instead of men and women recommend using males and females (since these are children).

·       Make NCEP column first since these parameters start at age 2, then put the IDF since they start at age 10.  Relabel adult panel—at what age does the adult panel apply? Instead of calling it the adult panel give the age range within your parameters.

·       If you don’t have parameters for 0 to age 3, suggest changing the age range from 3 to 24 years.

Line 69: Add source(s).

Line 79: Why did you opt to use an age range beyond 18 years?

Lines 123-128: Explain the scoring conventions for the two tools.

Lines 137-139: Remove directions.

Lines 325-328: Remove directions.

Line 329-330: Is this a true statement? How does this manuscript differ from the following journal article? McKay CD, O'Bryan E, Gubhaju L, McNamara B, Gibberd AJ, Azzopardi P, Eades S. Potential Determinants of Cardio-Metabolic Risk among Aboriginal and Torres Strait Islander Children and Adolescents: A Systematic Review. Int J Environ Res Public Health. 2022 Jul 27;19(15):9180

Supplemental Table 1:

·       It was hard on my computer to distinguish between the light gray and the white.

·       Recommend having all column lines visible. I find the current format confusing. Perhaps put Not Applicable in the columns with no data, or a line.

·       Add CREATE scores, too.

None

Author Response

  • Comment 1: while the two papers both investigate cardiometabolic risk in Aboriginal and Torres Strait Islander young people (aged 0–24 years), they differ substantially in their respective foci. Our paper reports the prevalence of cardiovascular risk markers in Aboriginal and Torres Strait Islander young people (0–24 years) in Australia, whereas McKay et al. investigated the association between potential determinants of health (e.g., large birth size, socioeconomic status) and these risk markers. The two papers therefore provide a holistic insight into both the prevalence of these risk markers and the potential causes driving their disproportionately high rates in Aboriginal and Torres Strait Islander young people. 
  • The age range of 0–24 years was chosen as it encompasses the period of childhood and youth as defined by the United Nations. Additional evidence and discussion of this has been included in the introduction. 
  • 'Top End' amended to Northern Australia to avoid confusion amongst international readers.
  • Comments 4–18: these errors have been amended in the manuscript.
  • Comment 19: as above
  • Comments 20–22: formatting has been changed to allow for easier interpretation of supplementary table 1

Reviewer 2 Report

·         Line 21: Abstract should start with some brief background information: A sentence or two giving a broad introduction to the study is required.

·         Line 22, 26: Is the adolescence age between 17 to 24 years of age? Is 24 the age of adolescence? This is confusing.

·   Line 26-27: Define these studies (cross-sectional, longitudinal, retrospective…etc.).

·         Line 45: Define this study.

·         Line 47: Delete (95% CI 6.0–7.4)- no need for numerical results.

·         Line 56: A longitudinal study, American native children?

·         Line 57: Delete (OR 11.5, 95% CI 2.1–63.7).

·         Line 58: young people age 0-19 years?

·         Figure 1: It would be good if authors could include the number of records through selected databases (i.e., Scopus=200, Embase=500…etc).

·         Figure 1 should be clear enough to read. For example, 201 Not population of interest or…….

·         Figure 2. Reason for exclusion- small sample size. Why? Please clarify. Few studies in S Table 1 were with small sample size.

·         The risk of bias assessment should be clarified in a separate table.

·         S Table 1: It is unclear how the quality was assessed (Good, fair, poor)?

·        Table 2: Why the term “youth” (15-24 years) was used? Please be consistent throughout.

·         I miss a discussion on gender differences in cardiometabolic risk markers.

·         Authors should expand on policy implications in the conclusions.

Author Response

  • Comment 1: additional information has been included to provide further context for the review. 
  • Comment 2: as with comments made by reviewer 1, additional evidence and discussion has been included to justify the age range of 0–24 years. 
  • Comments 3–8: manuscript amended to reflect this feedback. 
  • Comment 10: formatting amended.
  • Comment 11: as per the review's exclusion criteria, studies were excluded if sample size was less than 60 participants (n<60). 
  • Comment 13: quality was assessed using two screening tools, the Johanna Briggs Institute Critical AppraisaI Tool for prevalence studies and the The Centre of Research Excellence in Aboriginal Chronic Disease Knowledge Translation and Exchange Critical Appraisal Tool. The scores for these tools for each included study is provided in Supplementary Table 1. Two additional tables are included in the appendix that demonstrate the number of papers that met each item for the two respective criteria. 
  • Comment 14: terminology has been made consistent throughout the paper.
  • Comment 15: additional discussion on sex differences has been included.  
  • Comment 16: additional discussion on policy implications has been included. 

Reviewer 3 Report

The authors are to be congratulated for this comprehensive analysis of existing data on cardiometabolic risk in Aboriginal and Torres Island youth. One hopes these data will be helpful to policymakers and researchers into the health of marginalized cohorts. 

As a non-epidemeologist, I would ask the authors to define "grey literature" in the methods.

The definition for obesity in the youngest age group is not provided. Was this based on BMI-Z scores?

Please delete the first two lines of the discussion.

Author Response

  • Comment 2: a definition for grey literature has been included in the paper for your reference. 
  • Comment 3: the criteria used to define obesity in this study has been included.  
  • Comment 4: amended. 

Round 2

Reviewer 1 Report

The revision addresses the concerns I expressed during my initial review except for Table 1. The authors made edits that appear to be related to my comments; however, I believe a fourth column which provides metrics for 18-24 years is needed.

Recommend proofreading.

Author Response

Thank you for your feedback. Table 1 has been updated to include the International Diabetes Federation adolescent criteria for Metabolic Syndrome.

Reviewer 2 Report

No further comments.

Author Response

Thank you for your feedback.